# Divergence of the PIERCE1 expression between mice and humans as a p53 target gene

Hye Jeong Kim, Seung Eon Lee, Heeju Na, Jae-Seok Roe, Jae-il Roh ●*, Han-Woong Lee ●*

Department of Biochemistry, College of Life Science and Biotechnology, Yonsei University, Seoul, Republic of Korea

* rohjaeil@gmail.com (JIR); hwl@yonsei.ac.kr (HWL)

**Data Availability Statement:** All relevant data are within the manuscript and its Supporting Information files.

**Funding:** This work was supported by grants from the National Research Foundation of Korea (NRF;

## Abstract

PIERCE1, p53 induced expression 1 in Rb null cells, is a novel p53 target involved in the DNA damage response and cell cycle in mice. These facts prompted us to study the function of PIERCE1 with respect to p53-associated pathophysiology of cancer in humans. Unexpectedly, PIERCE1 did not respond to overexpression and activation of p53 in humans. In this study, we swapped p53 protein expression in human and mouse cells to find the clue of this difference between species. Human p53 expression in mouse cells upregulated *PIERCE1* expression, suggesting that p53-responsive elements on the *PIERCE1* promoter are crucial, but not the p53 protein itself. Indeed, *in silico* analyses of *PIERCE1* promoters revealed that p53-responsive elements identified in mice are not conserved in humans. Consistently, chromatin immunoprecipitation-sequencing (ChIP-seq) analyses confirmed p53 enrichment against the *PIERCE1* promoter region in mice, not in human cells. To complement the p53 study in mice, further promoter analyses suggested that the human *PIERCE1* promoter is more similar to guinea pigs, lemurs, and dogs than to rodents. Taken together, our results confirm the differential responsiveness of *PIERCE1* expression to p53 due to species differences in *PIERCE1* promoters. The results also show partial dissimilarity after p53 induction between mice and humans.

## Introduction

PIERCE1, p53-induced expression 1 in retinoblastoma (RB)-null cells also known as C9orf116 and RbEst47, was first shown to be upregulated in Rb-deficient mouse embryonic fibroblasts (MEFs) [1]. PIERCE1 is highly induced by p53 overexpression and activation because it consists of three p53-responsive elements at the region adjacent to the transcription start site [2]. As a result, genotoxic stresses such as ultraviolet C (UVC) light can activate PIERCE1 expression [2]. PIERCE1 expression fluctuates during the cell cycle, and its protein stability is highly affected by the proteasome system as it harbors the PEST domain at the N-terminus [1]. PIERCE1 expression is restricted in normal human tissues such as the brain, kidney, and lung

https://www.nrf.re.kr/eng/index) funded by the Korean government (MEST; 2018R1A2A1A05022746 and 2017R1A4A1015328) to Han-Woong Lee. The funders had no role in study design, data collection and analysis, decision to publish, or preparation of the manuscript.

**Competing interests:** The authors have declared that no competing interests exist.

[1]. *In vivo* analyses of PIERCE1 knockout (KO) mice revealed that approximately half of PIERCE1 homozygous KO mice exhibit embryonic lethality due to developmental defects, and 40% of survivors have *situs inversus totalis*, which reverses all organs from their normal positions [3].

p53, as an essential tumor suppressor gene, is a guardian of the genome that is the most frequently mutated or deleted in approximately 50% of all human cancers [4–7]. Activated p53 under stress conditions regulates target gene expression, resulting in the alteration of multiple physiologies including senescence, apoptosis, metabolism, and the cell cycle [8–11]. p53 is a sequence-specific pleiotropic transcription factor that binds specifically to a palindromic consensus sequence, RRRCWWGYYY ($N_{0-13}$) RRRCWWGYYY [12] with at least one mismatch. Mutation hotspots in p53 are mainly found in the DNA binding region across tumor types, suggesting that transcriptional activity is critical for its tumor suppressor function [13–15].

Since PIERCE1 is significantly induced after p53 overexpression and activation in mice, we examined tumor related PIERCE1 function under p53 activation and mutation conditions in humans. Unexpectedly, PIERCE1 expression was not affected by p53 overexpression and activation in normal and cancerous human cell types, while mouse cells successfully induced *Pierce1* expression regardless of p53 protein species origins. At the molecular level, the human *PIERCE1* promoter did not harbor the p53-responsive element near the transcription start site, but mice have three responsive elements. Though mice have been used as a typical model organism due to several genetic and physiological advantages and similarities to humans [16], recent reports suggested that mouse models are less reliable as representative human disease models [17]. Thus, as a part of p53 targets are differ between two species, molecular and biochemical data that comparing mice and humans need to be accounted more carefully especially in aspect to p53.

## Materials & methods

### Mice

FVB/NTac mice and p53 KO (Trp53^tm1Tyj^) mice were purchased from Taconic Biosciences and Jackson Laboratory, respectively [18]. p53 heterozygous KO mice were backcrossed onto the FVB/N for more than 20 generations. All mice were maintained in the specific pathogen-free (SPF) facility of the Yonsei Animal Research Center [19–21]. Animal suffering was minimized, and all experiments were performed in accordance with the Korean Food and Drug Administration (KFDA) guidelines for animal research. The experimental protocols for generating the genetically engineered mouse models were approved by the Institutional Animal Care and Use Committee (IACUC-A-201709-359-03) at Yonsei University.

### Cell culture

B16F1, H1299, and HCT-116 cells were purchased from the Korean Cell Line Bank (KCLB). MCF-7, A549, and BJ cells were purchased from the American Type Culture Collection (ATCC). Mouse embryonic fibroblasts (MEFs) were prepared and cultured as previously described [1]. Briefly, MEFs were isolated from 13.5-day embryos of wild-type (WT) and p53 KO mice. Lung cancer cells were isolated from tumor nodules of the KRAS^La2^ mouse model [22]. AC-16, Beas-2b, and 293A cells were provided by WJ Park (Gwangju Institute of Science and Technology) and HW Park (Yonsei University, Department of Biochemistry). All cells were cultured and maintained in RPMI 1640 (Gibco) or Dulbecco's modified Eagle's medium (DMEM; Gibco) supplemented with 10% fetal bovine serum (FBS; Sigma) and 1% penicillin/streptomycin (Gibco) at 37˚C in a humidified chamber containing 5% $CO_2$ [23].

## Transfection, RNA isolation, and real-time quantitative PCR

RNA was isolated and used for quantitative PCR as previously described [24]. In brief, total RNAs from cells were prepared with TRIsure reagent (Bioline) after treating for 12 hours with cisplatin (Sigma) or transfection. Cell extracts were harvested after 24 hours of transfection with Lipofectamine 3000 reagent (Invitrogen) or 36 hours of transfection with RNAiMAX reagent (Thermo Fisher Scientific). One microgram of total RNA was reverse-transcribed using the RevertAid First Strand cDNA Synthesis Kit (Thermo Scientific), and quantitative PCR was performed using SYBR Green (Bioline) and CFX Connect Real-Time System (Bio Rad) [25]. Experiments were conducted in triplicate, and signals were normalized to mouse and human *GAPDH*. All primers and siRNA sequences are listed in S1 and S2 Tables.

## Reporter gene constructs and luciferase assays

Reporter genes were constructed by subcloning the genomic DNA fragment containing the *PIERCE1* promoter region into the pGL3 Basic vector (Promega) [2]. p53 response elements were predicted by directly comparing DNA sequences with the consensus sequence [2, 12]. The mutant *PIERCE1* promoter constructs were generated using PCR-based mutagenesis [26] with the following primers: 5'– AAC AGG ACG CCG CCT TGC CGC AGC AGG CAC AGA CTT GAT CGC TTC TCC TCC AGG CAC AAT GT–3' and 5'– GAG GAG AAG CGA TCA AGT CTG TGC CTG CTG CGG CAA GGC GGC GTC CTG TTG CCA AGC GAC GG–3' for BS1; 5'– GGC AGG TTC CAG ACT TGC CTA CAG CTA GCT GCC CGG CCC ACG CGC GGC GCC TT–3' and 5'– GTG GGC CGG GCA GCT AGC TGT AGG CAA GTC TGG AAC CTG CCG GGC GAC TCC CCA AG–3' for BS2 and were verified by sequencing. Reporter genes and effector constructs pcDNA3-p53 were transfected with Lipofectamine 3000 reagent (Invitrogen). Luciferase assays were conducted as previously described [27].

## Chromatin immunoprecipitation-sequencing (ChIP-seq) enrichment peak analysis

The enrichment peaks were analyzed as previously described [28]. In brief, human and mouse p53 ChIP-seq data were aligned to the human and mouse reference genome assemblies hg19 and mm9 [29, 30], respectively. The peaks were re-calculated by normalizing to read depth of 10 million. The Gene Expression Omnibus accession number for the acquisition of publicly available ChIP-Seq data reported in this paper is GSE55727 (p53 ChIP-Seq in MEF) and GSE86164 (p53 ChIP-Seq in HCT116 and MCF7).

## Statistical analyses

Statistical significance was determined using the two-tailed t-test. Data were analyzed with GraphPad Prism (GraphPad Software). The value of relative difference was transformed into log 10. *P*-values less than 0.05 were considered statistically significant.

# Results

## Differential regulation of PIERCE1 expression by p53 in mice and humans

PIERCE1 is a novel p53 target involved in the DNA damage response in mice [1, 2]. Since mice are predicted to have similar genetic and physiological aspects to humans [31, 32], these data prompted us to examine whether *PIERCE1* expression is affected by p53 and apply the previous results to humans to investigate p53-related and tumor regulating functions of

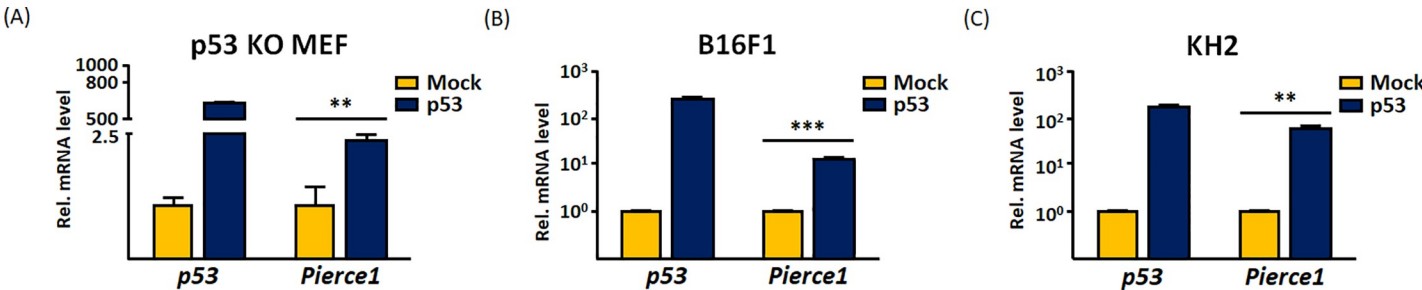

**Fig 1. p53-mediated *Pierce1* induction in mice.** (A–C) Relative *p53* and *Pierce1* mRNA level determined using qPCR in cells 24 hours after transient transfection with empty vector (yellow bars) or p53 (dark blue bars) in indicated mouse cell lines. p53-deficient mouse embryonic fibroblasts (MEFs, A) and cancer cells (B, melanoma; C, lung cancer) were used for the analyses. *GAPDH* was used for normalization. Data are presented as mean ± SD (Student's t-test, **$P < 0.01$, ***$P < 0.001$).

PIERCE1. Consistent with previous data, ectopic p53 expression significantly upregulated *Pierce1* expression approximately 2.5- and 10-fold in mouse primary fibroblasts and cancer cells, respectively (Fig 1) [2]. However, unlike p53-dependent regulation in mouse cells, the transcript level of *PIERCE1* was not affected by p53 overexpression in p53-deficient HCT-116 cells, even though induction of other p53 targets was confirmed (Fig 2A and 2B). Additionally, transient p53 overexpression and knockdown could not alter the *PIERCE1* expression in six different cell lines originating from humans (Fig 2C–2G), suggesting that *PIERCE1* is not the downstream target of p53 in humans.

Since genotoxic stress such as platinum treatment can promote p53 activity followed by induction of its downstream targets [6], genotoxic stresses were applied to examine gene expression alteration of p53 targets. Consistent with the previous report [2], significant upregulation of both *p21* and *Pierce1* was detected with cisplatin treatment conditions in MEFs (S1 Fig). However, unlike mice, the *PIERCE1* mRNA level was not changed by cisplatin in human primary fibroblast and cancer cell lines, but the *p21* mRNA level increased (Fig 3). These data implied that genotoxic stress-mediated p53 activation is not sufficient to induce human *PIERCE1*.

## Poor transcriptional activation is not due to p53 protein, but low conservation of p53 responsive elements in human PIERCE1 promoter

The differential responsiveness of *PIERCE1* expression to p53 can be due to either the p53 protein or *PIERCE1* promoter. To validate the factor that determines the species difference, the cDNA encoding human p53 was introduced into mouse cells and vice versa. *Pierce1* transcript was significantly upregulated up to 23.5-fold when human p53 was overexpressed in MEFs, the response which is similar to that of mouse p53 (Figs 1A and 4A), whereas overexpression of mouse p53 did not affect *PIERCE1* expression in p53 KO HCT-116 cells (Fig 4B and 4C). These results indicated that the *PIERCE1* promoter, rather than the p53 protein, is crucial for determining the species differences.

Since the *PIERCE1* promoter is predicted to differ between mice and humans, *in silico* analyses of the *PIERCE1* promoter were conducted to examine which element varies between them. Based on the previous reports [2, 12], the mouse *Pierce1* promoter was compared to the human promoter and promoters in other species, especially in the regions of p53-responsive elements (Fig 5A and S2 Fig). By re-analyzing publicly available mouse and human p53 ChIP--Seq, we confirmed that only mouse *PIERCE1* promoter selectively accumulates p53 while the promoter of known p53 target (i.e. *CDKN1A*) is occupied regardless of mouse and human cells (Fig 5B and 5C and S3 Fig). We measured p53 occupancy at the *PIERCE1* promoter and

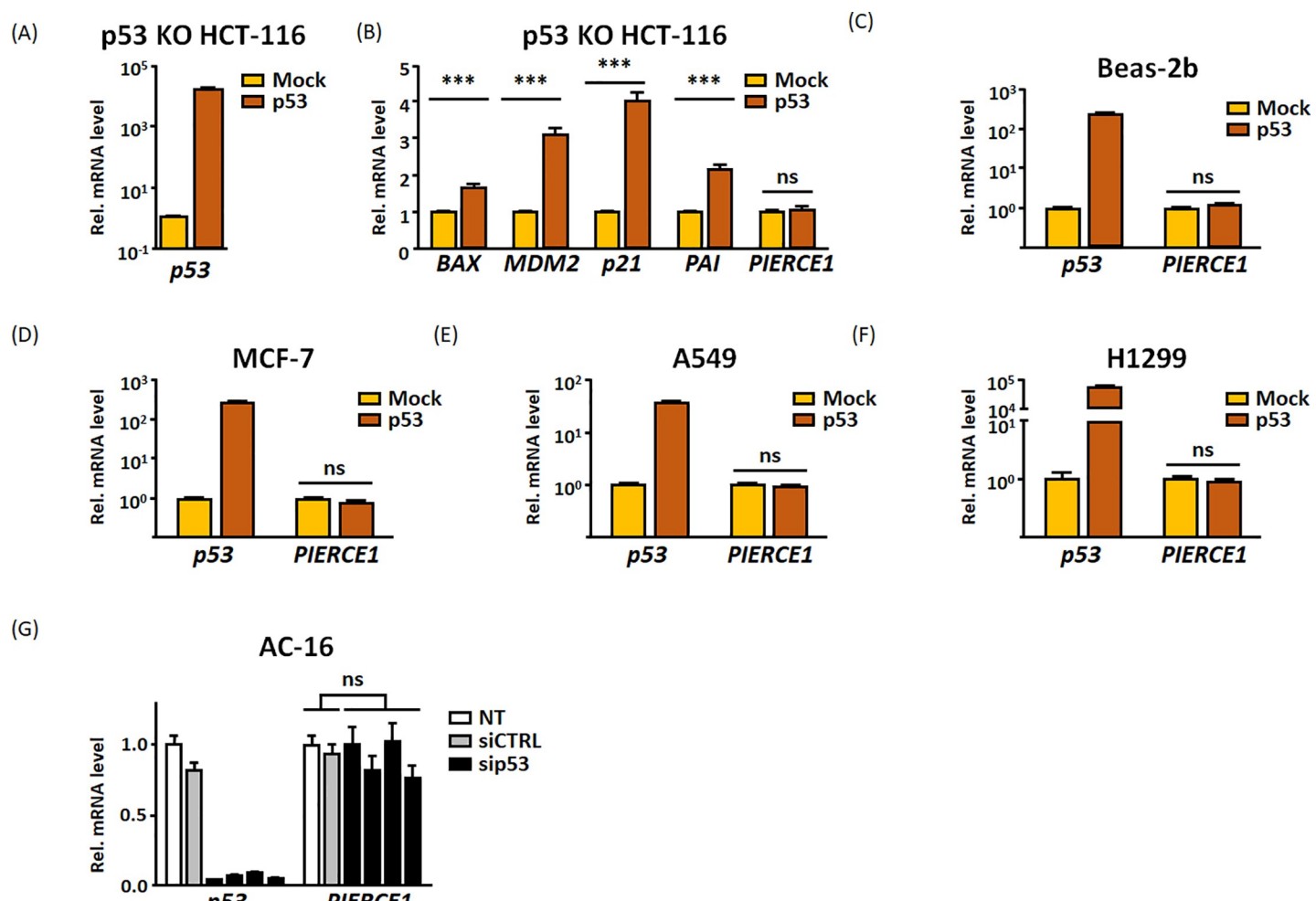

**Fig 2. PIERCE1 expression is not controlled by p53 in humans.** (A-B) RT-PCR analyses of *p53* (A) and its target (B) mRNA levels 24 hours after transient transfection of empty vector (yellow bars) and p53 (red bars) in p53 KO HCT-116 cells. (C-F) RT-PCR analyses of *p53* and *PIERCE1* transcripts 24 hours after transient transfection of control (yellow bars) and p53 (red bars) in Beas-2b (**C**), MCF-7 (D), A549 (E), and H1299 (F) cell lines. (G) Relative *p53* and *PIERCE1* mRNA levels of no transfection (NT, white bars) or 36 hours after transient transfection of siRNA against control (siCTRL, gray bars) or p53 (sip53, black bars) in AC16 cells. The level of each gene was normalized to *GAPDH*. *ns* indicates no significant difference. Data are presented as mean ± SD (Student's t-test, ***$P<0.001$).

compared it to the well-known p53 target gene, *CDKN1A*, in MEFs, HCT-116, and MCF-7 cell lines. Given that the p53 activation status is based on *CDKN1A* peaks, significant binding affinity at *Pierce1* promoter was identified in MEFs. No remarkable interaction was seen in HCT-116, and MCF-7 cells (Fig 5B and 5C). These data indicated that p53 binding sites (BSs) in *PIERCE1* promoter are not conserved in humans.

To investigate whether alteration of putative p53 BSs from the human to mouse sequences can affect p53 responsiveness, we used mutant forms of human *PIERCE1* promoter constructs called BS1mut, BS2mut, and BS1/2mut, which contain either mouse p53 binding site 1 or 2 (BS1 and 2), or both in human *PIERCE1* promoter, respectively (Fig 5A and 5D). As expected, luciferase analyses revealed no notable activation of human *PIERCE1* promoter by p53. However, mutant promoters, especially the constructs containing the BS1 mutation, showed significant upregulation of promoter activity in response to p53 (Fig 5D). Consistent with previous

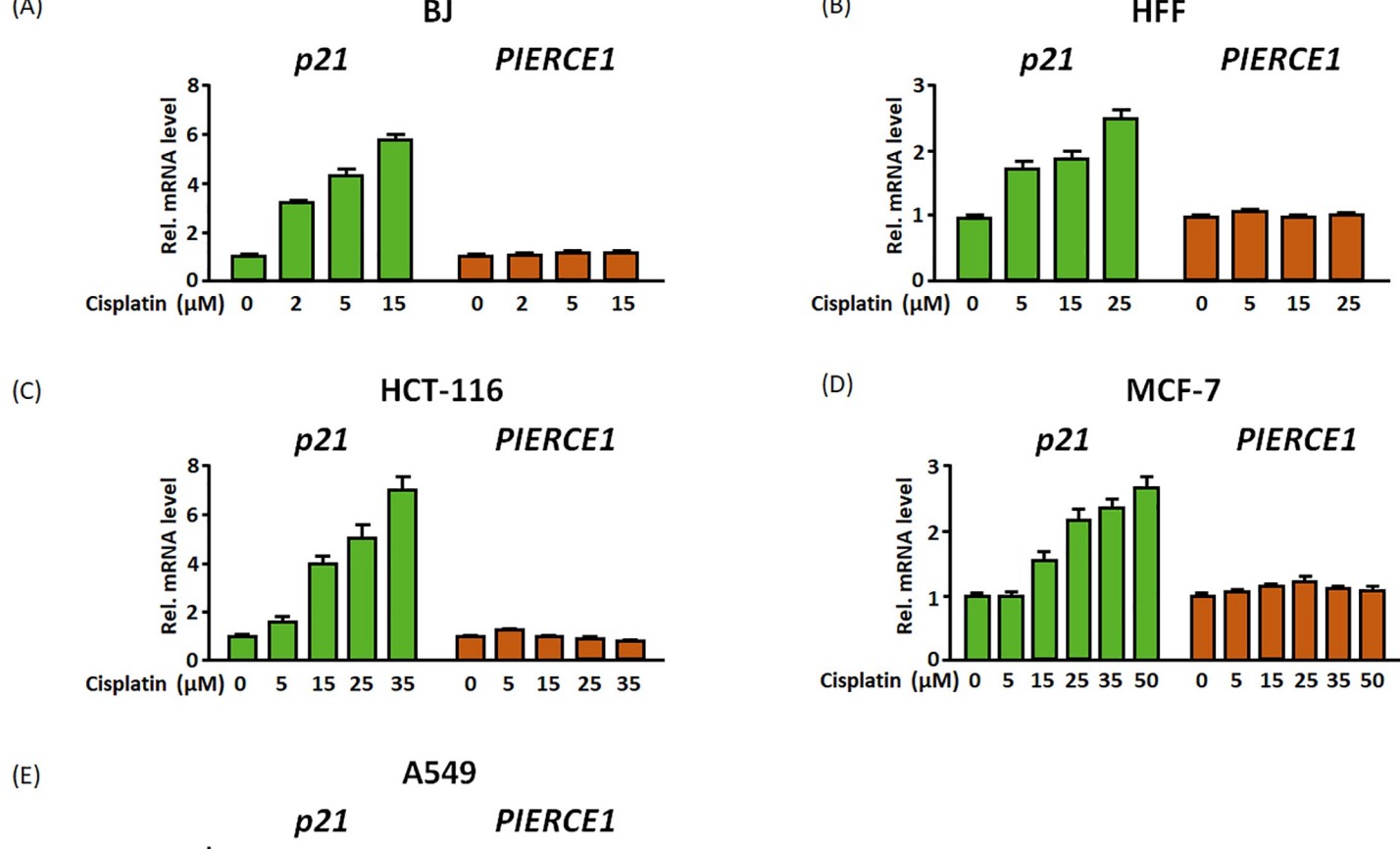

**Fig 3. *PIERCE1* expression is not induced by cisplatin in human cells.** (A–E) Relative expression of *p21* (green bars) and *PIERCE1* (red bars) transcripts after 12 hours of cisplatin treatment at the indicated doses in BJ (A), HFF (B), p53 KO HCT-116 (C), MCF-7 (D), and A549 (E) cell lines. *GAPDH* was used for normalization. Data are presented as mean ± SD (Student's t-test, triplicate).

data [2], BS2mut did not respond to p53. These data suggest that p53-response elements are the determining factors of PIERCE1 expression across the species.

## Proposal of alternative experimental animal models for p53 study

Although the mouse is the most frequently used model to examine p53 function, our data suggested that there is some inconsistency between mouse data and human cases. Thus, we investigated the *PIERCE1* promoter in the representative animal models to suggest representatives with more similarity to humans. Sequence analyses in the *PIERCE1* promoter region revealed that rat has high p53 BS consensus sequence homology with mouse, but these sites are poorly conserved in primates such as chimpanzees and humans (Fig 5A, the data for chimpanzee are not shown). Furthermore, the *PIERCE1* promoter phylogenetic tree analyses suggested that zebrafish, dog, pika, and guinea pig have similar PIERCE1 promoter structure to humans (S2

(A)

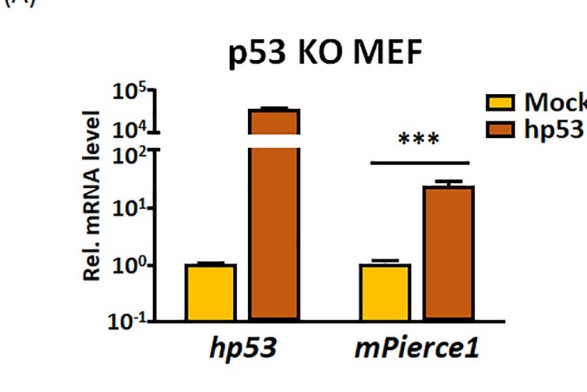

(B)                                                      (C)

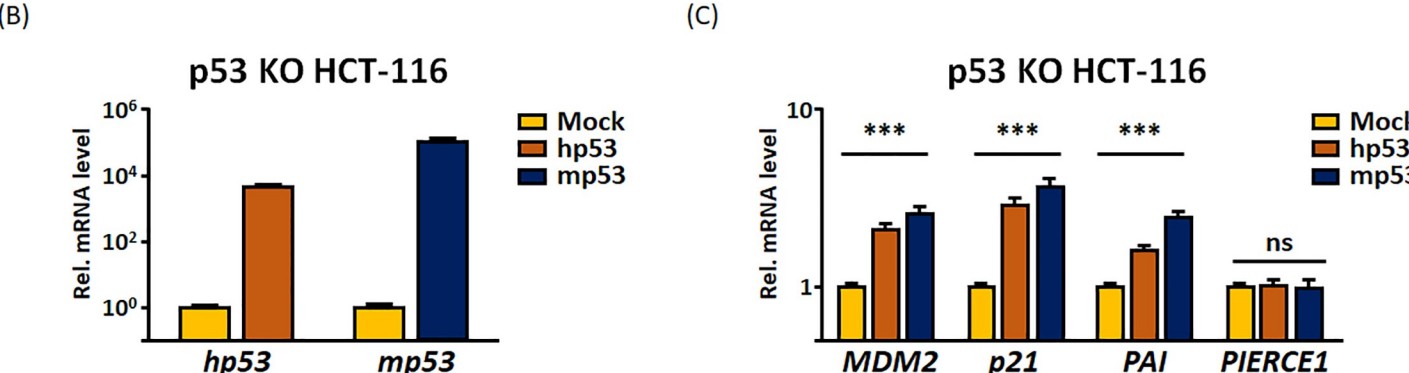

**Fig 4. Species differences in the *PIERCE1* promoter and not in the p53 protein is the key factor for *PIERCE1* induction by p53.** (A) Relative mRNA level of *p53* and *Pierce1* 24 hours after transient transfection of empty vector (yellow) or human p53 (red) in p53 KO MEF. (B-C) Relative mRNA level of *p53* (B) and its targets (C) 24 hours after transient expression of empty vector (yellow bars), human p53 (red bars), and mouse p53 (dark blue bars) in p53 KO HCT-116 cells. *GAPDH* was used for normalization. Data are presented as mean ± SD (Student's t-test, ***$P<0.001$; ns, no significance).

Fig). Additionally, previous reports indicated that *Lpin1* and *Cpt1c* genes are differentially controlled between mice and humans [33]. We also found that p53-responsive elements in these genes are preserved in rats, partially in guinea pigs and lemurs, but not in dogs and primates (S4 Fig). Furthermore, this tendency was stronger in the more closely related species. Collectively, these data indicate that mouse p53 is not fully representative of humans. On the contrary, guinea pig, lemur, and dog models might show a better correlation with human p53 function.

## Discussion

Here, we identified PIERCE1 as a differentially-regulated gene across species due to its promoter region variation. These dissimilarities inhibit the responsiveness of PIERCE1 to human p53. In a proof-of-concept, mutagenesis of the human PIERCE1 promoter at p53-responsive elements to the mouse allele successfully enabled transcriptional activation of the human PIERCE1 by p53 or genotoxic stress conditions.

Analyses of PIERCE1 promoter sequences in several species suggested that zebrafish, dogs, guinea pigs, and lemurs were predicted to be similar to humans in expression patterns and p53 or genotoxic stress responsiveness. On the contrary, transcriptional activity and responsiveness to p53 in the mouse *Pierce1* promoter might be similar to rats, frogs, squirrels, and tree shrews. The transcriptional responses to p53 are not restricted in PIERCE1, but approximately

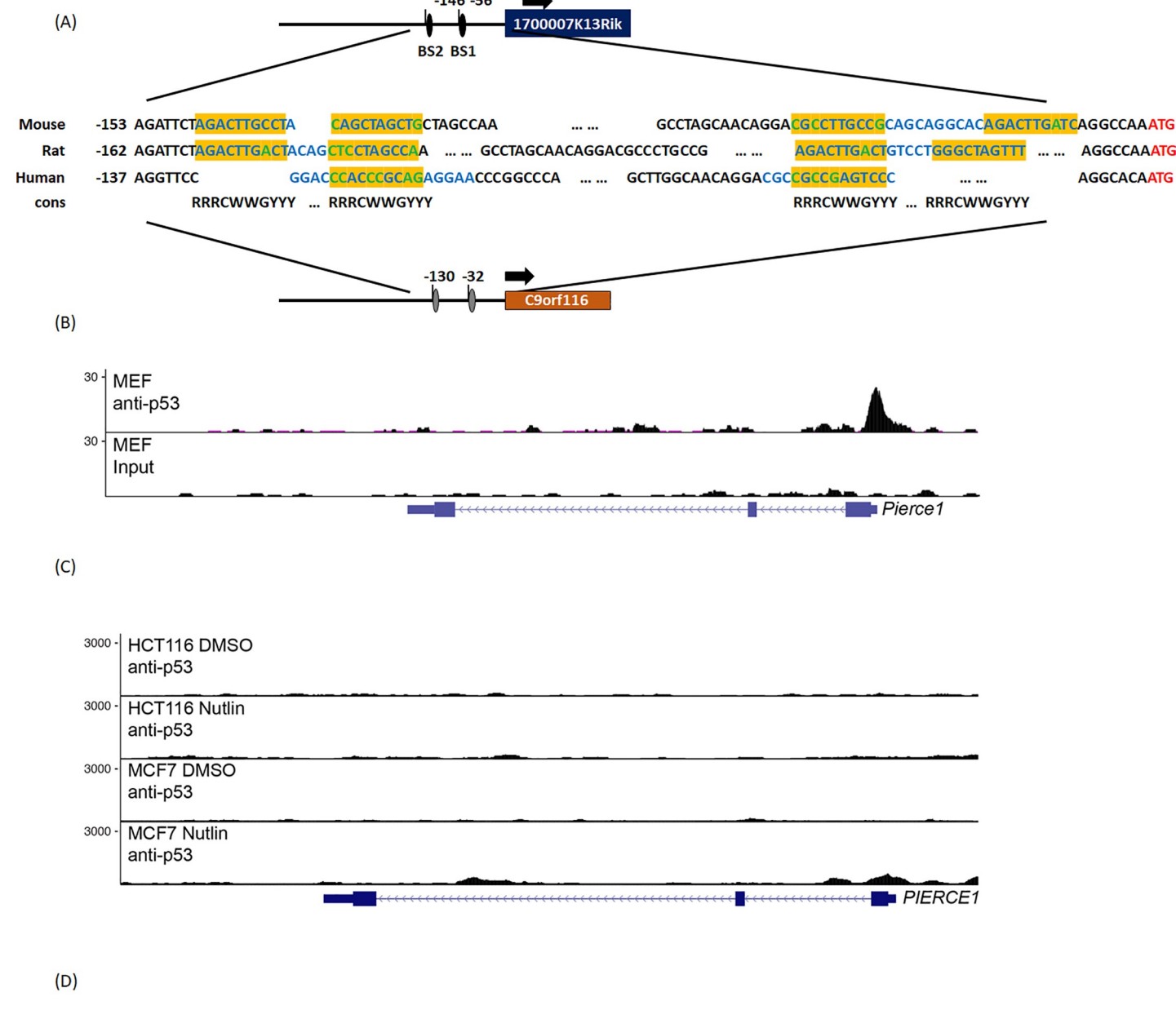

**Fig 5. *PIERCE1* promoter is the determining factor of varying p53 responsiveness in mice and humans.** (A) A schematic view of *PIERCE1* promoter region with p53-responsive elements (BS, shown in blue) in the mouse *Pierce1* locus (upper panel) and predicted BS sites in the *PIERCE1* locus of human and rat (human locus is shown in the bottom panel). Predicted BS sites are shown in blue, and two RRRCWWGYYY sequences, are highlighted in yellow. Mismatches with the *p53* consensus sequence are shown in green. R, purine; Y, pyrimidine; W, adenine, or thymine. Arrows indicate the transcription start region. (B-C) ChIP-sequencing profiles and identified peaks of direct p53 binding to *PIERCE1* in mouse cell, MEFs, (B) and human cell lines HCT-116, and MCF-7 (C). A schematic view below the profiles shows the *PIERCE1* gene and promoter region. Exons are shown as blue boxes, and introns are marked by blue lines. (D) A conceptual diagram of WT and mutant *PIERCE1* promoters with putative p53 BSs and relative luciferase activity of these promoters 24 hours after transfection of empty vector (yellow bars) or p53 (red bars) in 293A cells. Each mutant *PIERCE1* constructs contains mouse p53 binding site as described with ovals. Gray bars, human *PIERCE1* promoter; White ovals, WT; Black ovals, p53 BS sequence of mouse *Pierce1*. Data are presented as mean ± SD (Student's t-test, triplicate).

1010 genes including *POLK* [34], *LPIN1* [35], and *CPT1C* [36] also show species variation in their expression [33]. Gene ontology (GO) classification of these genes suggested that they are related to the DNA damage response, DNA metabolism, and cell cycle [33]. p53-induced DNA damage response and cell cycle alteration could vary between mice and humans due to the dissimilar expression of p53-dependent downstream target genes. Therefore, the use of animal models showing differential p53 responsiveness might interrupt proper p53 function interpretation.

## Supporting information

**S1 Fig. *Pierce1* expression is responsible for cisplatin-induced p53 activation in mice.** Relative mRNA level of *Pierce1* 12 hours after cisplatin treatment at the indicated doses in MEFs. GAPDH was used for normalization. Data are presented as mean ± SD (Student's t-test, triplicate).
(TIF)

**S2 Fig. Phylogenetic tree of the p53 binding sites of the PIERCE1 promoter region in the indicated species.** Deep blue lines show that no binding site exists. Light blue and yellow lines are predicted to harbor only BS2 or BS1, respectively. Red lines indicate that both two binding sites exist. (BS, binding site; O, exist; X, does not exist).
(TIF)

**S3 Fig. Identification of direct p53 binding sites.** ChIP-seq profiles and identified peaks of direct p53 binding to *CDKN1A* in mouse cells (MEFs) (**A**) and human HCT-116 and MCF-7 cell lines (**B**). The *CDKN1A* gene schematic is shown below. Exons are shown as blue boxes, and introns are marked in blue lines.
(TIF)

**S4 Fig. p53 responsiveness prediction based on promoter sequence analysis in several species.** p53 response sequence alignment of the *Lpin1* (**A**) and *Cpt1c* (**B**) in mice, and its consensus regions in rats, guinea pigs, lemurs, dogs, and humans. Predicted response elements are shown in blue, and two RRRCWWGYYY sequences separated by 0 to 13 bp of spacer DNA are highlighted in yellow. Mismatches with the p53 consensus sequences are shown in green. R, purine; Y, pyrimidine; W, adenine, or thymine.
(TIF)

**S1 Table. Primer sets for qPCR.**
(DOCX)

**S2 Table. siRNA sequences for p53 knockdown.**
(DOCX)

## Author Contributions

**Conceptualization:** Hye Jeong Kim, Jae-il Roh.

**Data curation:** Hye Jeong Kim.

**Formal analysis:** Hye Jeong Kim.

**Funding acquisition:** Han-Woong Lee.

**Investigation:** Hye Jeong Kim, Heeju Na, Jae-Seok Roe.

**Methodology:** Hye Jeong Kim, Heeju Na, Jae-Seok Roe.

**Project administration:** Jae-il Roh, Han-Woong Lee.

**Resources:** Jae-il Roh, Han-Woong Lee.

**Supervision:** Han-Woong Lee.

**Validation:** Seung Eon Lee, Heeju Na.

**Visualization:** Hye Jeong Kim, Heeju Na, Jae-Seok Roe.

**Writing – original draft:** Hye Jeong Kim, Jae-Seok Roe, Jae-il Roh, Han-Woong Lee.

**Writing – review & editing:** Hye Jeong Kim, Jae-Seok Roe, Jae-il Roh, Han-Woong Lee.

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
