## [Decision Letter · Decision Letter 0]

3 Apr 2020

PONE-D-20-03213

Divergence of the PIERCE1 expression between mice and humans as a p53 target gene

PLOS ONE

Dear Dr. Lee,

Thank you for submitting your manuscript to PLOS ONE. After careful consideration, we feel that it has merit but does not fully meet PLOS ONE’s publication criteria as it currently stands. Therefore, we invite you to submit a revised version of the manuscript that addresses the points raised during the review process.

Please respond to all critique, point-by-point. In particular:

- the  text would greatly profit from a  language checkup.

- page 8, line 164: Why is the induction by the human p53 protein not statistically significant?

- the reporter constructs should be explainded in more detail.

We would appreciate receiving your revised manuscript by May 18 2020 11:59PM. To enhance the reproducibility of your results, we recommend that if applicable you deposit your laboratory protocols in protocols.io, where a protocol can be assigned its own identifier (DOI) such that it can be cited independently in the future. For instructions see: http://journals.plos.org/plosone/s/submission-guidelines#loc-laboratory-protocols

We look forward to receiving your revised manuscript.

Kind regards,

Klaus Roemer

Academic Editor

PLOS ONE

Journal Requirements:

Reviewers' comments:

Reviewer's Responses to Questions

**Comments to the Author**

1. Is the manuscript technically sound, and do the data support the conclusions?

Reviewer #1: Yes

2. Has the statistical analysis been performed appropriately and rigorously? 

Reviewer #1: Yes

3. Have the authors made all data underlying the findings in their manuscript fully available?

Reviewer #1: Yes

4. Is the manuscript presented in an intelligible fashion and written in standard English?

Reviewer #1: Yes

5. Review Comments to the Author

Reviewer #1: Mice are often used as models for human diseases and so they are also used for cancer research. However, we human beings are not mice and this becomes particularly clear with studies like this one.

In the past, the authors have identified PIERCE1 as a DNA damage response and p53 target gene in mice. In a follow up study, the authors tried to transfer these results to the human system to investigate the impact of PIERCE1 for human cancer. However, they were unable to see an induction of PIERCE1 by p53 or DNA damage in the human system. They therefore decided to look deeper into this disparity. They started their analysis by swapping p53 expression constructs and cell lines originating from mouse and man and could observe that human p53 was able to induce PIERCE1 expression in mice but mouse p53 could not induce PIERCE1 in human cell lines while it was able to do so in murine cells, indicating that the PIERCE promoter rather than the p53 protein itself was responsible for the observed differences between murine and human cells. The authors then analysed the PIERCE1 promoter of different species and found that an important part of the p53 consensus sequence is missing in the PIERCE1 promoter of human cells. In silico analysis of p53-ChIP-seq data and supporting wet lab experiments confirmed a presence of p53 at the PIERCE1 promoter in murine cells and its absence in human cells. Finally, the authors analysed reporters that contained parts of the PIERCE1 promoter region either from human or murine cells.

This is a very clear story and I have therefore only minor comments:

- The abstract is a bit confusing. It also contains two spelling mistakes. Maybe the authors could try to write the abstract in a way that makes it better understandable also for people outside the field.

- Title figure 1: In all the text, PIERCE is written in big letters, only here in small ones. Please be consistent.

- Title figure 3: please make it more simple (e.g. PIERCE1 is not induced by cisplatin in human cells)

- Please label supplementary figures in the order in which they appear in the text

- There are several spelling/grammatical errors in the text

- Page 8; lane 164/165: This is not a similar degree. The human p53 seems to induce PIERCE1 to a greater extent. Why is the induction by the human protein not statistically significant?

- Title figure 4: The title is quite complex. Why not something easy like: Species differences in the PIERCE1 promoter and not in the p53 protein is the key factor for PIERCE1 induction by p53

- Page 9, lane 185 “p53 BSs”: Please introduce abbreviations

- Page 10, figure 5D: please better explain the different reporter constructs

6. PLOS authors have the option to publish the peer review history of their article (what does this mean?). If published, this will include your full peer review and any attached files.

Reviewer #1: No

---

## [Author Response · Author response to Decision Letter 0]

11 Jul 2020

Dear Editor-in-Chief,

We thank the editor and reviewer for the constructive suggestions. We have prepared this document to address all comments raised by the editor and reviewer. We hope that our response is clear and accurate, improving the quality of our manuscript. All changes made in the manuscript can be traced through the tracking section, and they are also stated here in the response section. 

1. The abstract is a bit confusing. It also contains two spelling mistakes. Maybe the authors could try to write the abstract in a way that makes it better understandable also for people outside the field.

Thank you for pointing out our mistakes. We have precisely checked the manuscript and corrected all typos. We have also revised the contents of the manuscript to make it better comprehensive according to the advice of the reviewer. All changes can be traced through the tracking section.

2. Title figure 1: In all the text, PIERCE is written in big letters, only here in small ones. Please be consistent.

According to standard nomenclature guidelines (http://www.informatics.jax.org/mgihome/nomen/gene.shtml#gnas), mice gene symbols should be beginning with an uppercase letter, followed by lowercase letters. To follow the guideline, mRNA of mouse PIERCE1 is written in small letters.

3. Title figure 3: please make it more simple (e.g. PIERCE1 is not induced by cisplatin in human cells)

We thank to the reviewer’s suggestion.

Page 8, lane 158: We replaced the title “PIERCE1 expression is not dependent on cisplatin-induced p53 activation in humans.” by “PIERCE1 expression is not induced by cisplatin in human cells.”.

4. Please label supplementary figures in the order in which they appear in the text

The order of supplementary figures 3 and 4 is swapped as recommended by the reviewer as follow;

Page 9, lane 190: We replaced “S4 Fig” by “S3 Fig”.

Page 11, lane 235: We replaced “S3 Fig” by “S4 Fig”.

Page 13, “S3 Fig” section replaced by “S4 Fig”, and moved from lane 266 - 271 to lane 276 – 281. Also “S4 Fig” replaced by “S3 Fig” in lane 272.

The indicated correction has been also made inside these supplementary figures.

5. There are several spelling/grammatical errors in the text

Page 8, lane 165: We replaced the title “Poor transcription activation is…” by “Poor transcriptional activation is…”.

Page 10, lane 207: Gene symbols are italicized as “PIERCE1 promoter”

Other corrections can be traced through tracking section.

We also edited the manuscript with the help of a professional English editing service (Editage by Cactus Communications Inc., NJ 08540, USA).

6. Page 8; lane 164/165: This is not a similar degree. The human p53 seems to induce PIERCE1 to a greater extent. Why is the induction by the human protein not statistically significant?

We agree that the word “similar degree” can be confusable. As the reviewer mentioned, there is an approximately 10-fold difference between Pierce1 expression level induced by mouse and human p53. The original intention was that human p53 “also” induces expression of mouse Pierce1 in MEF as mouse p53. The revised sentence as follows will convey the meaning more precisely: 

Page 8, lane 170-171: We replaced “which is a similar degree of induction by mouse p53 (Fig 4A)” by “the response which is similar to that of mouse p53 (Fig 1A and Fig 4A)”.

7. Title figure 4: The title is quite complex. Why not something easy like: Species differences in the PIERCE1 promoter and not in the p53 protein is the key factor for PIERCE1 induction by p53

We agree with the reviewer’s comments and thank you for suggesting proper example. 

Page 9, lane 175: We replaced the title “Species differences in cell, but not p53 origin, is the key factor for PIERCE1 expression.” by “Species differences in the PIERCE1 promoter and not in the p53 proteins are the key factor for PIERCE1 induction by p53.”.

8. Page 9, lane 185 “p53 BSs”: Please introduce abbreviations

It stands for ‘binding sites’, which p53 directly binds to. We replaced the word ‘BSs’ in the manuscript as follow:

Page 9, lane 194-195: We replaced “p53 BSs” by “p53 binding sites (BSs)”.

9. Page 10, figure 5D: please better explain the different reporter constructs

The detailed explanation of mutant reporter constructs is added in the legends as follows:

Page 10, lane 209-210: We added, “Each mutant PIERCE1 constructs contains mouse p53 binding sites as described as ovals.”.

---

## [Editor Report · Decision Letter 1]

16 Jul 2020

Divergence of the PIERCE1 expression between mice and humans as a p53 target gene

PONE-D-20-03213R1

Dear Dr. Lee,

We’re pleased to inform you that your manuscript has been judged scientifically suitable for publication and will be formally accepted for publication once it meets all outstanding technical requirements.

Kind regards,

Klaus Roemer

Academic Editor

PLOS ONE
---

## [Editor Report · Acceptance letter]

20 Jul 2020

PONE-D-20-03213R1 

Divergence of the PIERCE1 expression between mice and humans as a p53 target gene 

Dear Dr. Lee:

I'm pleased to inform you that your manuscript has been deemed suitable for publication in PLOS ONE. Congratulations! Your manuscript is now with our production department. 

Kind regards, 

on behalf of

Dr. Klaus Roemer 

Academic Editor

PLOS ONE